# Real-Time Adjustment and Spatial Data Integration Algorithms Combining Total Station and GNSS Surveys with an Earth Gravity Model

Krzysztof Karsznia [1,*], Edward Osada [2] and Zbigniew Muszyński [3]

1. Faculty of Geodesy and Cartography, Warsaw University of Technology, Plac Politechniki 1, 00-661 Warszawa, Poland
2. Faculty of Technical Sciences, University of Lower Silesia, Wagonowa St. 9, 53-609 Wrocław, Poland; edward.osada@dsw.edu.pl
3. Faculty of Geoengineering, Mining and Geology, Wrocław University of Science and Technology, Wyb. Wyspiańskiego 27, 50-370 Wrocław, Poland; zbigniew.muszynski@pwr.edu.pl
* Correspondence: krzysztof.karsznia@pw.edu.pl

**Featured Application: The authors have developed a spatial data-fusion algorithm to adjust and effectively integrate data sets from total station and GNSS surveys with a universal EGM geoid model. The solution's high effectiveness can be applied to structural monitoring, risk management, architectural designing, and building information modelling (BIM).**

**Abstract:** During the dynamic development of modern technologies based on advanced algorithmic and instrumental solutions, it is essential to integrate geospatial data efficiently. Such an approach is applied in all geo-information services, especially mobile ones, and is helpful in, for example, precise navigation or effective risk management. One leading application is deformation monitoring (structural monitoring) and displacement control surveying. In addition, spatial data integration methods are used in modern accessibility analysis, Smart City ideas, tracing utility networks, and building information modelling (BIM). The last aforementioned technology plays a crucial role in architectural design and construction. In this context, it is crucial to develop efficient and accurate algorithms supporting data fusion, which do not strain the computing resources and operate efficiently online. This paper proposes an algorithm for real-time adjustment of integrated satellite GNSS (global navigation satellite system), total station, and Earth Gravitational Model (EGM) vertical direction data in a geocentric coordinate system based on a statistical general linear mixed model. A numerical example shows that the proposed algorithm of the online adjustment works correctly. The results of the online adjustment are the same as those of the offline adjustment. It is also shown that the GNSS measurements are necessary only at the total station points in the spatial total station traverse. There is no need to add additional merging points of the total station positions because the differences between the results of the online adjustment, including and excluding the merging points, are very small (around 1–2 mm in standard deviation).

**Keywords:** algorithms; data adjustment; data integration; integrated geodesy; structural monitoring

## 1. Introduction

Integrating geospatial measurements is essential in modern information services and geoportals using precise navigation, inventory registries of technical infrastructure, and studying the ongoing deformations of engineering objects. Such geo-information platforms are now fundamental when conducting construction works on, e.g., subway lines, highway tunnels, or other technical facilities. They enable effective risk management and structure the work in progress. Nowadays, almost every civil investment uses geoportals and advanced algorithms, allowing for real-time data fusion. Such effective management

of spatial data has increasingly been involving mobile technologies. In such cases, the activities of system analysts focus on the operation of smartphone or tablet devices, which requires the optimization of algorithmic solutions, thereby ensuring efficient switching between functions.

In the case of structural monitoring systems, notifications of potential threats in the form of predefined alarm thresholds are generated based on previously captured and properly integrated data. The user can receive such alerts in an ongoing form, except for simple visualization on screens, also employing signals, text messages, or sound calls. Reliable generation of the warnings mentioned above is required because property and often human lives depend on the speed of response to appearing circumstances. Moreover, such systems often use artificial intelligence and machine learning to initiate and aid appropriate reactions. The topic currently belongs to the mature systems and is reflected in many publications. For example, articles [1–4] present an overview of existing solutions in this field, explaining the theoretical basis in detail. The cited works also include an in-depth analysis of the available literature, considering new solutions and presenting them as complementary. In the context of spatial data acquisition, photogrammetric techniques—mainly images taken using UAVs (unmanned aerial vehicles) and close-range photogrammetry [5]—are used to a remarkable extent. Data-driven applications are used in mobile solutions, open visions, or complex services, e.g., Smart City concepts [6]. In this context, it is worth mentioning the so-called Internet of Things (IoT), through which multiple sources, especially mobile data, can be acquired and processed online [7,8].

However, this article focuses on analytical solutions to present a mechanism for effectively integrating spatial data obtained from different sensors, particularly geodetic observables. Such an approach makes it possible to visualize and interpret the survey results with millimeter accuracy. That, in turn, makes it possible to conduct assessments of geometric changes in engineering structures while examining their displacements and defining related risk management [9]. Numerous publications have also described research on the development of computational algorithms and data adjustment in integrated geodesy. Such works date even back to the late 1980s and early 1990s, for example [10,11]. In one author's solution, which was the topic of a doctoral dissertation and the results of which are presented in the article [12], a methodology was developed for the adjustment and integration of spatial traverses surveyed using total stationtechniques with reference to GNSS satellite measurements and the geoid. The measurement experiment was conducted both in urban areas and in the mountains. The total stationand GNSS results were further enriched by modelling the mutual plumb line arrangement in successive observation stations. Various polynomial functions were tested for this purpose, and numerical algorithms were assessed, including Levenberg–Marquardt's [13]. The use of numerical methods in landslide monitoring is also described in the work [14]. Depending on the type of landslide under study, an appropriate interpolation grid was defined in the test area, distinguishing sectors of the phenomenon's low, medium, and high activity. The level of activity then determines the selection of appropriate modelling parameters. The method was developed utilizing the total station data; however, it can be modified depending on other geodetic measurement technologies. For example, such integration can be performed by employing photogrammetric methods of correlating digital images [15].

Research on integrating total station and GNSS satellite measurements is also a vibrant subject amongst researchers. For example, the publication [16] presents and discusses study results using the methods above for deformation surveys of a large dam, obtaining promising results in the long-term evaluation of its technical condition. Article [17], on the other hand, demonstrates the analysis of a mathematical model for the adjustment and integration of geodetic measurements in the local spatial system, without the need for prior projection of GNSS vectors onto a reference ellipsoid. The author proposed a computational model that integrates data from two sources—total station and static GNSS measurements—and their joint adjustment in a single, coherent coordinate system. However, this approach considers any local geoid model. In addition, the implementation

of GNSS measurements requires a maximally open horizon, and in the case of other than static positioning methods (e.g., Real-Time), also the fulfillment other conditions such as an adequate number of available satellites, reducing interfering elements (multipath mitigation), and many more [18].

As previously mentioned, photogrammetric methods based on appropriate modelling of the relative and absolute positions of image projection centers while ensuring adequate coverage of the pictured object [19] are currently the subject of exciting research. Based on many available studies in this area, the authors conducted research employing photogrammetry in precise object dimensioning and structural monitoring [20]. In this case, however, one can talk about pure local network-specific micro-grids with reference points deployed directly on the examined object. Similar studies can also be found in [21]. Here, for assessing the condition of historical objects, the authors proposed the integration of multi-source photogrammetric data, including historical photographs. In this context, the publication [22] is also noteworthy, in which the authors proposed using geomatics technologies for the digital reconstruction of historical objects currently undergoing renewal.

The above considerations motivated the authors to conduct studies on developing computational methods for integrating multi-source data processing, significantly increasing the accuracy and reliability of the results obtained. Due to the integration of GNSS measurements given directly in the geocentric reference frame and the vertical directions from the EGM model (also computed in the geocentric reference frame), this frame is used in the joint adjustment of the total station and GNSS measurements.

The effects of the relevant research are summarized and organized as follows: the literature review is provided in the Introduction, the theoretical basis of the conducted research is presented in Materials and Methods and its subsequent sections, and the results of experimental works are further discussed (Discussion) and concluded (Conclusions), showing some propositions for further research.

## 2. Materials and Methods

Based on theoretical assumptions, in this section, we present the workflow of the developed method.

### 2.1. General Assumptions

Three-dimensional models of terrestrial objects are built based on a set of points with coordinates $(X, Y, Z)$ obtained using direct (total station measurements, 3D laser scanning) and indirect methods (close-range photogrammetry, remote sensing). The accuracy of the constructed model depends on the quality of the coordinates obtained. In the case of direct methods, the accuracy of a measured point $Q$ (Figure 1) depends on sound knowledge of six total station or laser scanner external orientation parameters:

- $X_s, Y_s, Z_s$, the geocentric GRS80 coordinates of the origin $P$ of the total stationtotal station or a terrestrial laser scanner (TLS) measuring frame $(x, y, z)$;
- $\Sigma, \xi, \eta$, the orientation angles of the $(x, y, z)$ measuring frame with respect to the external reference frame $(X, Y, Z)$.

The $\xi$ and $\eta$ orientation angles are the components of the total station or TLS vertical axis deflection from the normal to the GRS80 ellipsoid. The directional horizontal angle S is called the instrument orientation constant.

The coordinates $x, y, z$ of the point Q measured by the total station can be expressed using the known form [23,24]:

$$x = d \cdot \cos \alpha \cdot \sin \beta \qquad (1)$$

$$y = d \cdot \sin \alpha \cdot \sin \beta \qquad (2)$$

$$z = d \cdot \cos \beta + i - j \qquad (3)$$

where d–spatial distance, $\alpha$–horizontal direction, $\beta$–vertical angle (corrected due to refraction), i–the height of the total station above ground point P; and j–the reflector (surveying prism)'s height above ground point Q.

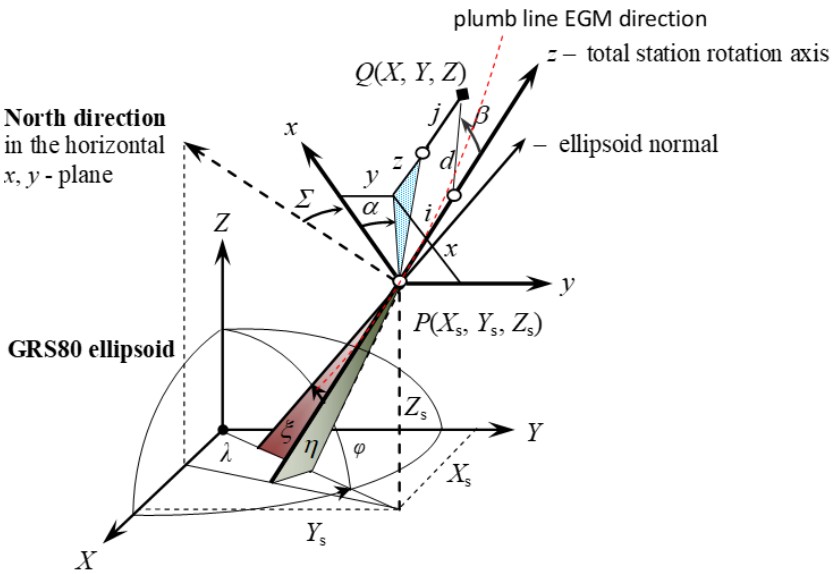

**Figure 1.** Geocentric vs. topocentric (instrumental) reference system: $X_s, Y_s, Z_s, \Sigma, \xi, \eta$ are six parameters of the total station external orientation; $\varphi, \lambda$ are geodetic latitude and longitude; $\alpha, \beta$ are measured horizontal and vertical angles to the point $Q(X, Y, Z)$; $d$ is the measured distance to the surveyed point $Q$; and $i, j$ are the instrument and reflector heights.

The coordinates $(x, y, z)$ are converted to the $(X, Y, Z)$ external reference system of the point Q, according to the well-known formula [25]:

$$\begin{pmatrix} X \\ Y \\ Z \end{pmatrix} = \begin{pmatrix} X_s \\ Y_s \\ Z_s \end{pmatrix} + (\mathbf{R}(\Sigma)\mathbf{Q}(\xi, \eta, \varphi)\mathbf{P}(\varphi, \lambda))^T \cdot \begin{pmatrix} x \\ y \\ z \end{pmatrix} \tag{4}$$

where

$$\mathbf{P}(\varphi, \lambda) = \begin{pmatrix} -\sin\varphi\cos\lambda & -\sin\varphi\sin\lambda & \cos\varphi \\ -\sin\lambda & \cos\lambda & 0 \\ \cos\varphi\cos\lambda & \cos\varphi\sin\lambda & \sin\varphi \end{pmatrix} \tag{5}$$

$$\mathbf{Q}(\xi, \eta, \phi) = \begin{pmatrix} 1 & -\eta\tan\phi & -\xi \\ \eta\tan\phi & 1 & -\eta \\ \xi & \eta & 1 \end{pmatrix} \tag{6}$$

$$\mathbf{R}(\Sigma) = \begin{pmatrix} \cos\Sigma & \sin\Sigma & 0 \\ -\sin\Sigma & \cos\Sigma & 0 \\ 0 & 0 & 1 \end{pmatrix} \tag{7}$$

The five external total station orientation parameters $(X_s, Y_s, Z_s, \xi, \eta)$ can be obtained from GNSS measurements $(X_s, Y_s, Z_s)$ and computed from the EGM gravity model [26]. The sixth parameter $S$ can be determined by the solution of Equation (4) for the given total station measurements $(d, \alpha, \beta, i, j)$ and the coordinates from GNSS measurements at the total station point $P(X_s, Y_s, Z_s)$ and the target point $Q(X, Y, Z)$ using, e.g., the Levenberg–Marquardt method of conjugate gradients [27].

The integrated total station/GNSS/EGM geocentric points' positioning method can be applied to measurements on a single total station position, as well as on a few merged total station positions along a spatial traverse (Figure 2).

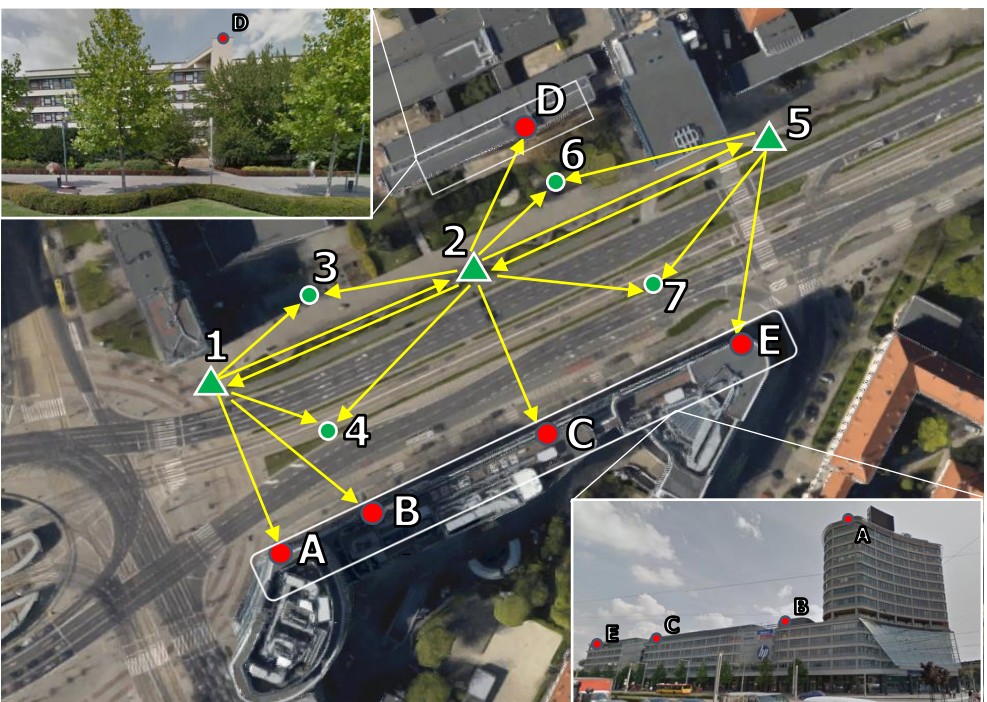

**Figure 2.** The total station spatial traverse points: the station points 1, 2, 5, the station merging points 3, 4, 6, 7, and the measured control points A, B, C, D, E, deployed on office buildings in Wroclaw (Poland): arrow pointers indicate the measured spatial directions (d, α, β) (picture by E. Osada, background: Google Maps). The station and merging points 1–7 are also measured using the GNSS-RTK technique.

The offline adjustment algorithm and the significance of the deflection of the vertical components $(\xi, \eta)$ for the ultimate adjustment results are widely discussed in prior papers [28–30]. In the current work, we focus on the online procedures of adjusting the total station spatial traverse in real time. The GNSS and EGM data are used for spatial orientation of the total station traverse with respect to the geocentric reference frame. The online adjustment methods presented in this paper are based on the general linear–mixed statistical model, e.g., [31]. It is defined based on linearized observational equations of the integrated data, as explained in the following Section 2.2. Alternative formulations of the real-time sequential adjustment algorithm can be found in papers showing applications of the Kalman filter and sequential regularization methods, e.g., [9,32], and in the case of the Moore–Penrose pseudo-inverse in [33].

*2.2. The Observational Equations of the Integrated Data*

The linear observational equation of the coordinates vector determined using total station measurements (formulas (1)–(3)) is given by

$$\begin{bmatrix} x + \varepsilon_x \\ y + \varepsilon_y \\ z + \varepsilon_z \end{bmatrix} = \begin{bmatrix} x_0 + dx \\ y_0 + dy \\ z_0 + dz \end{bmatrix} \tag{8}$$

Hence,

$$\begin{bmatrix} \varepsilon_x \\ \varepsilon_y \\ \varepsilon_z \end{bmatrix} = \begin{bmatrix} dx \\ dy \\ dz \end{bmatrix} - \begin{bmatrix} x - x_s \\ y - y_s \\ z - z_s \end{bmatrix} \tag{9}$$

where

- $dx, dy, dz$ are differentials of the coordinates $x, y, z$;

- $\varepsilon_x, \varepsilon_y, \varepsilon_z$ are the measurement random errors with zero expected value and known covariance matrix:

$$\begin{bmatrix} \sigma_x^2 & \sigma_{xy} & \sigma_{xz} \\ \sigma_{xy} & \sigma_y^2 & \sigma_{yz} \\ \sigma_{xz} & \sigma_{yz} & \sigma_z^2 \end{bmatrix} \tag{10}$$

- $x_s, y_s, z_s$ are approximate values of the measured coordinates:

$$\begin{bmatrix} x_s \\ y_s \\ z_s \end{bmatrix} = \mathbf{R}(\Sigma)\mathbf{Q}(\xi, \eta, \varphi)\mathbf{P}(\varphi, \lambda) \cdot \begin{bmatrix} X_c - X \\ Y_c - Y \\ Z_c - Z \end{bmatrix} \tag{11}$$

The elements of the covariance matrix (10), which stand for the functions of the direct observables (1)–(3), are computed using the covariance propagation law [34]:

$$\sigma_x = \sqrt{\sin^2 \beta \cos^2 \alpha \cdot \sigma_d^2 + d^2 \cos^2 \beta \cos^2 \alpha \cdot \sigma_\beta^2 + d^2 \sin^2 \beta \sin^2 \alpha \cdot \sigma_\alpha^2} \tag{12}$$

$$\sigma_y = \sqrt{\sin^2 \beta \sin^2 \alpha \cdot \sigma_d^2 + d^2 \cos^2 \beta \sin^2 \alpha \cdot \sigma_\beta^2 + d^2 \sin^2 \beta \cos^2 \alpha \cdot \sigma_\alpha^2} \tag{13}$$

$$\sigma_z = \sqrt{\cos^2 \beta \cdot \sigma_d^2 + d^2 \cdot \sin^2 \beta \cdot \sigma_\beta^2 + \sigma_i^2 + \sigma_j^2} \tag{14}$$

$$\sigma_{xy} = \frac{1}{2} \sin 2\alpha \cdot \left( \sin^2 \beta \cdot \sigma_d^2 + d^2 \cos^2 \beta \cdot \sigma_\beta^2 - d^2 \cdot \sin^2 \beta \cdot \sigma_\alpha^2 \right) \tag{15}$$

$$\sigma_{xz} = \sin \beta \cos \beta \cos \alpha \cdot \left( \sigma_d^2 - d^2 \cdot \sigma_\beta^2 \right) \tag{16}$$

$$\sigma_{yz} = \sin \beta \cos \beta \sin \alpha \cdot \left( \sigma_d^2 - d^2 \cdot \sigma_\beta^2 \right) \tag{17}$$

where $\sigma_d, \sigma_\alpha, \sigma_\beta, \sigma_i, \sigma_j$ are the standard deviations of the directly measured values $d, \alpha, \beta, i, j$.
The differential change of the total station coordinates equal the following:

$$\begin{bmatrix} x \\ y \\ z \end{bmatrix} = \mathbf{R}(\Sigma)\mathbf{Q}(\xi, \eta, \varphi)\mathbf{P}(\varphi, \lambda) \cdot \begin{bmatrix} X_c - X \\ Y_c - Y \\ Z_c - Z \end{bmatrix} \tag{18}$$

due to differential changes of the total station standpoint coordinates $X, Y, Z$, the target point coordinates $X_c, Y_c, Z_c$ and total station spatial orientation angles $\Sigma, \xi, \eta$ are given by

$$\begin{aligned}
\begin{pmatrix} dx \\ dy \\ dz \end{pmatrix} &= \mathbf{R}(\Sigma)\mathbf{Q}(\xi, \eta, \phi)\mathbf{P}(\phi, \lambda) \cdot \begin{bmatrix} 1 & & \\ & 1 & \\ & & 1 \end{bmatrix} \begin{pmatrix} dX_c - dX \\ dY_c - dY \\ dZ_c - dZ \end{pmatrix} + \mathbf{R}(\Sigma)\frac{\partial \mathbf{Q}(\xi, \eta, \phi)}{d\xi}\mathbf{P}(\phi, \lambda)\begin{pmatrix} X_c - X \\ Y_c - Y \\ Z_c - Z \end{pmatrix}d\xi \\
&+ \mathbf{R}(\Sigma)\frac{\partial \mathbf{Q}(\xi, \eta, \phi)}{d\eta}\mathbf{P}(\phi, \lambda)\begin{pmatrix} X_c - X \\ Y_c - Y \\ Z_c - Z \end{pmatrix}d\eta + \frac{\partial \mathbf{R}(\Sigma)}{\partial \Sigma}\mathbf{Q}(\xi, \eta, \phi)\mathbf{P}(\phi, \lambda)\begin{pmatrix} X_c - X \\ Y_c - Y \\ Z_c - Z \end{pmatrix}d\Sigma \\
&+ \mathbf{R}(\Sigma)\frac{\partial [\mathbf{Q}(\xi, \eta, \phi)\mathbf{P}(\phi, \lambda)]}{d\phi}\begin{pmatrix} X_c - X \\ Y_c - Y \\ Z_c - Z \end{pmatrix}d\phi + \mathbf{R}(\Sigma)\frac{\partial [\mathbf{Q}(\xi, \eta, \phi)\mathbf{P}(\phi, \lambda)]}{d\lambda}\begin{pmatrix} X_c - X \\ Y_c - Y \\ Z_c - Z \end{pmatrix}d\lambda
\end{aligned} \tag{19}$$

where the differentials $d\varphi, d\lambda$ are functions of the differentials $dX, dY, dZ$ [35]:

$$\begin{pmatrix} d\phi \\ d\lambda \\ dh \end{pmatrix} = \begin{bmatrix} \frac{1}{M+h} & 0 & 0 \\ 0 & \frac{1}{(N+h)\cos\phi} & 0 \\ 0 & 0 & 1 \end{bmatrix} \mathbf{P}(\phi, \lambda) \begin{pmatrix} dX \\ dY \\ dZ \end{pmatrix} \tag{20}$$

and $M, N$ are the radii of curvature of the ellipsoid in the meridian and in the prime vertical, respectively.

The rotation $(\Sigma, \xi, \eta)$ of the total station reference frame $(x, y, z)$ due to differential displacement $d\varphi, d\lambda$ of the total station point is very small and can be neglected. Thus, the terms containing differentials $d\varphi, d\lambda$ can be omitted, and Equation (19) is reduced to

$$
\begin{pmatrix} dx \\ dy \\ dz \end{pmatrix} = \mathbf{R}(\Sigma)\mathbf{Q}(\xi, \eta, \phi)\mathbf{P}(\phi, \lambda) \begin{bmatrix} 1 & & \\ & 1 & \\ & & 1 \end{bmatrix} \begin{pmatrix} dX_c - dX \\ dY_c - dY \\ dZ_c - dZ \end{pmatrix} + \mathbf{R}(\Sigma)\frac{\partial \mathbf{Q}(\xi, \eta, \phi)}{d\xi} \begin{pmatrix} x_g \\ y_g \\ z_g \end{pmatrix} d\xi
$$
$$
+\mathbf{R}(\Sigma)\frac{\partial \mathbf{Q}(\xi, \eta, \phi)}{d\eta} \begin{pmatrix} x_g \\ y_g \\ z_g \end{pmatrix} d\eta + \frac{\partial \mathbf{R}(\Sigma)}{\partial \Sigma}\mathbf{Q}(\xi, \eta, \phi) \begin{pmatrix} x_g \\ y_g \\ z_g \end{pmatrix} d\Sigma
\tag{21}
$$

where

$$
\begin{pmatrix} x_g \\ y_g \\ z_g \end{pmatrix} = \mathbf{P}(\varphi, \lambda) \begin{pmatrix} X_c - X \\ Y_c - Y \\ Z_c - Z \end{pmatrix}
\tag{22}
$$

and

$$
\frac{\partial \mathbf{Q}(\xi, \eta, \varphi)}{d\xi} = \begin{bmatrix} 0 & 0 & -1 \\ 0 & 0 & 0 \\ 1 & 0 & 0 \end{bmatrix}
\tag{23}
$$

$$
\frac{\partial \mathbf{Q}(\xi, \eta, \varphi)}{d\eta} = \begin{bmatrix} 0 & -\tan\varphi & 0 \\ \tan\varphi & 0 & -1 \\ 0 & 1 & 0 \end{bmatrix}
\tag{24}
$$

$$
\frac{\partial \mathbf{R}(\Sigma)}{\partial \Sigma} = \begin{bmatrix} -\sin\Sigma & \cos\Sigma & 0 \\ -\cos\Sigma & -\sin\Sigma & 0 \\ 0 & 0 & 0 \end{bmatrix}
\tag{25}
$$

Finally, the reduced linearized observational equation of the total station-measured coordinates $x, y, z$ (8) and (9) is given by the following formula:

$$
\begin{pmatrix} \varepsilon_x \\ \varepsilon_y \\ \varepsilon_z \end{pmatrix} = \begin{bmatrix} -\sin\lambda\sin\Sigma - \sin\phi\cos\lambda\cos\Sigma & \cos\lambda\sin\Sigma - \sin\phi\sin\lambda\cos\Sigma & \cos\phi\cos\Sigma \\ -\sin\lambda\cos\Sigma + \sin\phi\cos\lambda\sin\Sigma & \cos\lambda\cos\Sigma + \sin\phi\sin\lambda\sin\Sigma & -\cos\phi\sin\Sigma \\ \cos\phi\cos\lambda & \cos\phi\sin\lambda & \sin\phi \end{bmatrix} \begin{pmatrix} dX_c - dX \\ dY_c - dY \\ dZ_c - dZ \end{pmatrix}
$$
$$
+ \begin{bmatrix} -z_g\cos\Sigma & x_g\sin\Sigma\tan\phi - y_g\cos\Sigma\tan\phi - z_g\sin\Sigma & -x_g\sin\Sigma + y_g\cos\Sigma \\ z_g\sin\Sigma & x_g\cos\Sigma\tan\phi + y_g\sin\Sigma\tan\phi - z_g\cos\Sigma & -x_g\cos\Sigma - y_g\sin\Sigma \\ x_g & y_g & 0 \end{bmatrix} \begin{bmatrix} d\xi \\ d\eta \\ d\Sigma \end{bmatrix} - \begin{pmatrix} x - x_0 \\ y - y_0 \\ z - z_0 \end{pmatrix}
\tag{26}
$$

The linear observational equation of the vector of geocentric coordinates $X, Y, Z$ (obtained from GNSS measurements) is given by

$$
\begin{pmatrix} X + \varepsilon_X \\ Y + \varepsilon_Y \\ Z + \varepsilon_Z \end{pmatrix} = \begin{pmatrix} X_s + dX \\ Y_s + dY \\ Z_s + dZ \end{pmatrix} \Rightarrow \begin{pmatrix} \varepsilon_X \\ \varepsilon_Y \\ \varepsilon_Z \end{pmatrix} = \begin{pmatrix} dX \\ dY \\ dZ \end{pmatrix} - \begin{pmatrix} X - X_0 \\ Y - Y_0 \\ Z - Z_0 \end{pmatrix}
\tag{27}
$$

where $dX, dY, dZ$ are corrections of the coordinates $X, Y, Z$; $X_s, Y_s, Z_s$ are their approximate values, and $\varepsilon_x, \varepsilon_y, \varepsilon_z$ are measurements random errors with zero expected value and a known covariance matrix.

$$
\begin{bmatrix} \sigma_X^2 & \sigma_{XY} & \sigma_{XZ} \\ \sigma_{XY} & \sigma_Y^2 & \sigma_{YZ} \\ \sigma_{XZ} & \sigma_{YZ} & \sigma_Z^2 \end{bmatrix}
\tag{28}
$$

The linear observational equations of the vertical deflection components $\xi, \eta$ are given by

$$\begin{pmatrix} \xi + \varepsilon_\xi \\ \eta + \varepsilon_\eta \end{pmatrix} = \begin{pmatrix} \xi_0 + d\xi \\ \eta_0 + d\eta \end{pmatrix} \Rightarrow \begin{pmatrix} \varepsilon_\xi \\ \varepsilon_\eta \end{pmatrix} = \begin{pmatrix} d\xi \\ d\eta \end{pmatrix} - \begin{pmatrix} \xi - \xi_0 \\ \eta - \eta_0 \end{pmatrix} \tag{29}$$

where $d\xi, d\eta$ are corrections of the vertical component deflection $\xi, \eta$; $\xi_0, \eta_0$ are their approximate values, and $\varepsilon_\xi, \varepsilon_\eta$ are measurement random errors with zero expected value and a covariance matrix:

$$\begin{bmatrix} \sigma_\xi^2 & \sigma_{\xi\eta} \\ \sigma_{\xi\eta} & \sigma_\eta^2 \end{bmatrix} \tag{30}$$

*2.3. Adjustment of the Integrated Data at the Current Position of the Total Station Using the Statistical General Linear Mixed Model*

The statistical general linear mixed model (e.g., [31]) for all possible data measured at the current total station position $(d, \alpha, \beta, X, Y, Z, \xi, \eta)$, represented by linear observational Equations (8)–(17), including corrections to all the parameters adjusted at the previous total station position, is defined as

$$\varepsilon = \mathbf{X}\boldsymbol{\beta} + \mathbf{U}\boldsymbol{\gamma} - \mathbf{y} \tag{31}$$

where

$$\begin{bmatrix} \boldsymbol{\gamma} \\ \boldsymbol{\varepsilon} \end{bmatrix} \sim \left( \begin{bmatrix} \mathbf{0} \\ \mathbf{0} \end{bmatrix}, \begin{bmatrix} \boldsymbol{\Sigma}_\gamma & \mathbf{0} \\ \mathbf{0} & \boldsymbol{\Sigma}_\varepsilon \end{bmatrix} \right) \tag{32}$$

In this model, X and U are known design matrices, y is a known data vector, $\beta$ is a vector of unknown corrections to all new parameters included at the current position of the total station, $\boldsymbol{\gamma} \sim (\mathbf{0}, \boldsymbol{\Sigma}_\gamma)$ is a vector of unknown corrections to all parameters adjusted at previous total-station positions with zero expected value, and the known covariance matrix $\boldsymbol{\Sigma}_\gamma$, $\boldsymbol{\varepsilon} \sim (\mathbf{0}, \boldsymbol{\Sigma}_\varepsilon)$ is a vector of unknown observational residuals with zero-expected value and a known covariance matrix $\boldsymbol{\Sigma}_\varepsilon$.

The least squares solution $\boldsymbol{\varepsilon}\boldsymbol{\Sigma}_\varepsilon^{-1}\boldsymbol{\varepsilon}^{\mathbf{T}} + \boldsymbol{\gamma}\boldsymbol{\Sigma}_\gamma^{-1}\boldsymbol{\gamma}^{\mathbf{T}} = \min$ of the mixed model $\boldsymbol{\varepsilon} = \mathbf{X}\boldsymbol{\beta} + \mathbf{U}\boldsymbol{\gamma} - \mathbf{y}$ (31) is given by [31]

$$\hat{\boldsymbol{\beta}} = \left( \mathbf{X}^T \boldsymbol{\Sigma_y}^{-1} \mathbf{X} \right)^{-1} \mathbf{X}^T \boldsymbol{\Sigma_y}^{-1} \mathbf{y} \tag{33}$$

$$\hat{\boldsymbol{\gamma}} = \boldsymbol{\Sigma_\gamma} \mathbf{U}^T \boldsymbol{\Sigma_y}^{-1} (\mathbf{y} - \mathbf{X}\hat{\boldsymbol{\beta}}) \tag{34}$$

where

$$\boldsymbol{\Sigma_y} = \boldsymbol{\Sigma_\varepsilon} + \mathbf{U}\boldsymbol{\Sigma_\gamma}\mathbf{U}^T \tag{35}$$

Or, in a block matrix notation:

$$\begin{bmatrix} \hat{\boldsymbol{\beta}} \\ \hat{\boldsymbol{\gamma}} \end{bmatrix} = \begin{bmatrix} \boldsymbol{\Sigma}_{\hat{\beta}} & \boldsymbol{\Sigma}_{\hat{\beta}\hat{\gamma}} \\ \boldsymbol{\Sigma}_{\hat{\beta}\hat{\gamma}}^{\mathbf{T}} & \boldsymbol{\Sigma}_{\hat{\gamma}} \end{bmatrix} \begin{bmatrix} \mathbf{X}^T\boldsymbol{\Sigma}_\varepsilon^{-1}\mathbf{y} \\ \mathbf{U}^T\boldsymbol{\Sigma}_\varepsilon^{-1}\mathbf{y} \end{bmatrix} \tag{36}$$

where

$$\begin{bmatrix} \boldsymbol{\Sigma}_{\hat{\beta}} & \boldsymbol{\Sigma}_{\hat{\beta}\hat{\gamma}} \\ \boldsymbol{\Sigma}_{\hat{\beta}\hat{\gamma}}^{\mathbf{T}} & \boldsymbol{\Sigma}_{\hat{\gamma}} \end{bmatrix} = \begin{bmatrix} \mathbf{X}^T\boldsymbol{\Sigma}_\varepsilon^{-1}\mathbf{X} & \mathbf{X}^T\boldsymbol{\Sigma}_\varepsilon^{-1}\mathbf{U} \\ \mathbf{U}^T\boldsymbol{\Sigma}_\varepsilon^{-1}\mathbf{X} & \mathbf{U}^T\boldsymbol{\Sigma}_\varepsilon^{-1}\mathbf{U} + \boldsymbol{\Sigma}_\gamma^{-1} \end{bmatrix}^{-1} \tag{37}$$

$$\boldsymbol{\Sigma}_{\hat{\beta}} = \left( \mathbf{X}^T \boldsymbol{\Sigma_y}^{-1} \mathbf{X} \right)^{-1} \tag{38}$$

$$\boldsymbol{\Sigma}_{\hat{\beta}\hat{\gamma}} = -\boldsymbol{\Sigma}_{\hat{\beta}} \mathbf{X}^T \boldsymbol{\Sigma_y}^{-1} \mathbf{U} \boldsymbol{\Sigma_\gamma} \tag{39}$$

$$\boldsymbol{\Sigma}_{\hat{\gamma}} = \left( \mathbf{U}^T \boldsymbol{\Sigma}_\varepsilon^{-1} \mathbf{U} + \boldsymbol{\Sigma}_\gamma^{-1} \right)^{-1} - \boldsymbol{\Sigma}_{\hat{\beta}\hat{\gamma}}^T \mathbf{X}^T \boldsymbol{\Sigma_y}^{-1} \mathbf{U} \boldsymbol{\Sigma_\gamma} \tag{40}$$

is the covariance matrix of the adjusted parameters $\hat{\boldsymbol{\beta}}$ and $\hat{\boldsymbol{\gamma}}$.

### 2.4. Experimental Works

The experimental works were performed in Wrocław, Poland. The proposed algorithms were tested on data obtained during measurements at the points of the spatial traverse (Figure 2). At the total station traverse positions 1, 2, 5, there were spatial distances *s*, horizontal and vertical angles $\alpha$, $\beta$ (Figure 1), surveyed with standard deviations $\sigma_d$ = 0.006 m, $\sigma_\alpha = \sigma_\beta$ = 0.0010 grad, and the instrument height *i* and reflector (prism) height *j*, with $\sigma_i = \sigma_j$ = 0.002 m. Measurements were performed with the use of the Leica FlexLine TS02 total station [36]. The *X*, *Y*, *Z* coordinates of the station points 1, 2, 5 and merging points 3, 4, 6, 7 were measured with a Leica GS10 GNSS receiver [37] with standard deviations $\sigma_X = \sigma_Y = \sigma_Z$ = 0.008 m. At each point, we performed real-time kinematic (RTK) GNSS surveys with direct differential corrections broadcast by the GNSS permanent station WROC, located at the Wrocław University of Environmental and Life Sciences. The observation data can be downloaded from http://www.asgeupos.pl/index.php?wpg_type=syst_descr&sub=ref_st&st_id=wroc (accessed on 18 July 2023). The distance from the mobile receiver to the reference station WROC was less than 250 m.

The northern $\xi$ and eastern $\eta$ components of the deflection of the total station vertical axis from normal to the GRS80 ellipsoid were computed at the points 1, 2, 5 from the EGM2008 geoid model data, available from https://www.usna.edu/Users/oceano/pguth/md_help/html/egm96.htm (accessed on 18 July 2023) and presented in Table 1.

**Table 1.** Northern and eastern components of the plumb line deviations in the total station stand points (EGM2008), see Figure 1.

| Point No. | $\xi$-Component [arc sec $''$] | $\eta$-Component [arc sec $''$] | $\sum$ [grad] |
|:---:|:---:|:---:|:---:|
| 1 | 5.9926 | 6.2033 | 209.0750 |
| 2 | 5.9852 | 6.1967 | 296.7144 |
| 5 | 5.9775 | 6.1896 | 50.2572 |

The approximate value $\Sigma_1$ was computed by solving Equation (4) for GNSS-surveyed point coordinates *X*, *Y*, *Z* of the points 1 and 2, and *x*, *y*, *z* coordinates of the point 2 measured using total station set up in point 1. Analogically, we computed the values $\Sigma_2$ and $\Sigma_5$.

According to information from total station manufacturers, the direction of the vertical axis of an instrument is consistent with the real plumb line direction to the order of $0.5''$–$2.0''$ [arc seconds]. So, in the adjustment process, the standard deviations of the components of deflection of the vertical are assumed to be $1''$ ($\sigma_\xi = \sigma_\eta$ = 0.0003 [grad]). The covariance $\sigma_{\xi\eta}$ is unknown; it is assumed as 0 in the matrix (31). Research has shown [38] that in mountainous areas, in the Alps, the actual direction of the vertical can differ up to $3.5''$ from the direction of the vertical computed in the EGM2008 model. This error is similar to the accuracy of the vertical angles used in the experiment's total-station instrument ($3.0''$). In the case of unique high-accuracy measurements using total stations in mountain areas, the accuracy of the vertical deflection computed from EGM2008 model can be improved up to $0.8''$ using a correction computed from the numerical terrain model [38].

An alternative method of joint total station and GPS positioning with the use of digital terrain and gravity models is given in the paper [39]. In this case, the spherical model of the local vertical distribution is extended using local digital models of the terrain and gravity. Then, the residual field of the vertical deflection, based on spherical harmonic polynomials' expansion, is determined via real-time adjustment of the total station and GPS data.

A method of adjustment of the total station data in real-time, assuming the spherical distribution of the local vertical, is given in [33]. In this case, the relative position vectors and height differences measured with total station along a traverse are adjusted in real-time in three independent modes: spatial, planar, and height adjustment. If there are

some relative position GPS vectors, GPS ellipsoidal height differences, and leveling height differences, they are included in the adjusting processes at the current position of the total station. In the case of the height adjustment, it is assumed that the equipotential surfaces are spherical in the local area. The adjustments are made using a recursive formulation of the least squares method featuring the Moore–Penrose pseudoinverse.

In the method considered in this article, the accuracy and spatial orientation of the total station traverse can also be improved, including additional measured height differences between the traverse points using a leveling instrument. The relationship between ellipsoidal and leveling orthometric or normal heights, which has been studied and developed in many publications, e.g., [40–47], should be considered.

### 2.4.1. Classical Offline Adjustment

The classical offline adjustment of the total station traverse is performed, having completed all measurements. In our numerical experiment, it has been performed in two variants: by "including" and "excluding" the merging points 3, 4, 6, 7 (Figure 2). In the first case, the traverse consists of three stations (1, 2, and 5) and four merging points (3, 4, 6, and 7), with coordinates determined using a GNSS receiver, and five measured control points on the high buildings A, B, C, D, and E. In the second case, the traverse consists of only three station points (1, 2, and 5), as determined by the GNSS receiver, and five measured control points (A, B, C, D, and E).

The differences between the geocentric coordinates of all points obtained from the "including" and "excluding" computation modes are not greater than 4.2 mm, with a mean value of 1.0 mm and a standard deviation of $\pm 1.4$ mm. Comparing these with the accuracy of the total station and GNSS measurements, we can conclude that the merging points are not necessary for the spatial orientation of the total station traverse. As we can see, the spatial orientation is very well provided by only the GNSS-measured coordinates of the total station points 1, 2, and 5, and the deflections of the vertical obtained from the EGM model.

The algorithms and results of the classical offline adjustment obtained via the measured data in the same experiment shown in Figure 2 are presented in the article [28].

An alternative to the offline adjustment of all observations after making measurements is to adjust the observations online directly during measurements, in real-time, and sequentially.

In Section 2.4.2, we present the online algorithm for sequential adjustment of the sets of observations performed at successive total station positions.

In Section 2.4.3, we present the online algorithm for sequential adjustment of the direct observations performed at successive total station positions.

Online adjustment of observations provides real-time information about the accuracy of the currently measured point, and about improving the accuracy of all previously surveyed points. Thus, it is possible to conduct additional observations in real-time, and select new total station positions to increase the accuracy of the measured points. These are the main practical advantages of the real-time online sequential adjustment method, in comparison with the offline method.

### 2.4.2. Sequential Online Adjustments of Total Station Positions

At the first total station position over the point 1 (see Figures 2 and 3), the observational–mixed model is given by the equation

$$\varepsilon = \mathbf{X}\boldsymbol{\beta} - \mathbf{y} \tag{41}$$

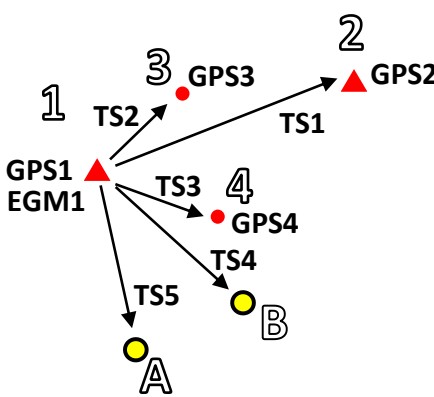

**Figure 3.** Surveying geometry at the total station position 1. The numbered points marked TS relate to total station surveys; GPS are measured using GNSS (GPS) technology, whereas EGM stands for the Earth Gravitational Model.

$$
\begin{bmatrix}
\boldsymbol{\varepsilon}_{\textbf{TS1}} \\
\boldsymbol{\varepsilon}_{\textbf{TS2}} \\
\boldsymbol{\varepsilon}_{\textbf{TS3}} \\
\boldsymbol{\varepsilon}_{\textbf{TS4}} \\
\boldsymbol{\varepsilon}_{\textbf{TS5}} \\
\hline
\boldsymbol{\varepsilon}_{\textbf{GPS1}} \\
\boldsymbol{\varepsilon}_{\textbf{GPS2}} \\
\boldsymbol{\varepsilon}_{\textbf{GPS3}} \\
\boldsymbol{\varepsilon}_{\textbf{GPS4}} \\
\hline
\boldsymbol{\varepsilon}_{\textbf{EGM1}}
\end{bmatrix}
= \mathbf{X}
\begin{bmatrix}
\mathbf{d}_1 \\
\mathbf{d}_2 \\
\mathbf{d}_3 \\
\mathbf{d}_4 \\
\mathbf{d}_A \\
\mathbf{d}_B \\
\hline
d\xi_1 \\
d\eta_1 \\
d\Sigma_1
\end{bmatrix}
- \mathbf{y}
\tag{42}
$$

where

- $\boldsymbol{\beta} = \begin{bmatrix} \mathbf{d}_1^T & \mathbf{d}_2^T & \mathbf{d}_3^T & \mathbf{d}_4^T & \mathbf{d}_A^T & \mathbf{d}_B^T & | & d\xi_1 & d\eta_1 & d\Sigma_1 \end{bmatrix}^T$ is the vector of corrections to the approximate geocentric coordinates $X, Y, Z$ of the points 1, 2, 3, 4, A, B (e.g., $\mathbf{d}_1^T = \begin{bmatrix} dX_1 & dY_1 & dZ_1 \end{bmatrix}$) and corrections $d\xi_1, d\eta_1, d\Sigma_1$ to the approximate orientation angles of the total station $\xi_1, \eta_1, \Sigma_1$,

- $\boldsymbol{\varepsilon} = \begin{bmatrix} \varepsilon_{TS1}^T & \varepsilon_{TS2}^T & \varepsilon_{TS3}^T & \varepsilon_{TS4}^T & \varepsilon_{TS5}^T & | & \varepsilon_{GPS1}^T & \varepsilon_{GPS2}^T & \varepsilon_{GPS3}^T & \varepsilon_{GPS4}^T & | & \varepsilon_{EGM1}^T \end{bmatrix}^T$ is the vector of total station measurement random errors $\varepsilon_x, \varepsilon_y, \varepsilon_z$ of the $x, y, z$ coordinates of the points 2, 3, 4, B, A, respectively (e.g., $\varepsilon_{TS1}^T = \begin{bmatrix} \varepsilon_{x_1} & \varepsilon_{y_1} & \varepsilon_{z_1} \end{bmatrix}$); the GNSS measurements' random errors $\varepsilon_X, \varepsilon_Y, \varepsilon_Z$ of the $X, Y, Z$ coordinates of the points 1, 2, 3, 4, respectively (e.g., $\varepsilon_{GPS1}^T = \begin{bmatrix} \varepsilon_{X_1} & \varepsilon_{Y_1} & \varepsilon_{Z_1} \end{bmatrix}$); and the random errors $\varepsilon_\xi, \varepsilon_\eta$ of the deflection of the vertical components $\xi_1, \eta_1$ at the first total station position (at point 1): $\varepsilon_{EGM1}^T = \begin{bmatrix} \varepsilon_{\xi_1} & \varepsilon_{\eta_1} \end{bmatrix}$.

- $\mathbf{X}, \mathbf{y}$ is the known design matrix and free terms' vector.

The solution $\hat{\boldsymbol{\beta}}, \boldsymbol{\Sigma}_{\hat{\boldsymbol{\beta}}}$ of the observational model $\boldsymbol{\varepsilon} = \mathbf{X}\boldsymbol{\beta} - \mathbf{y}$ is given by Equations (33)–(40).

In the next total station standpoint 2 (see Figures 2 and 4), the observational–mixed model $\boldsymbol{\varepsilon} = \mathbf{X}\boldsymbol{\beta} + \mathbf{U}\boldsymbol{\gamma} - \mathbf{y}$ is given by

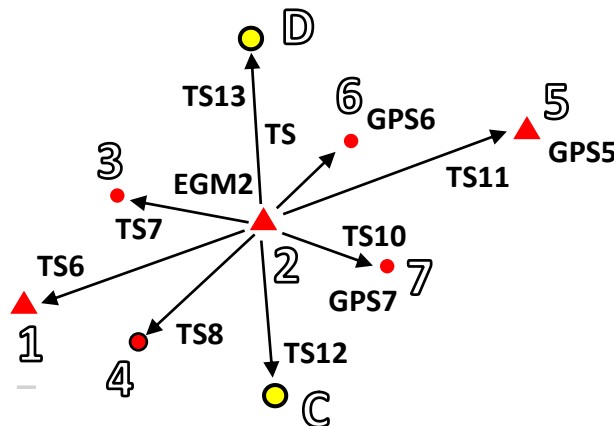

**Figure 4.** Surveying geometry at the total station position 2. The numbered points marked as TS relate to total station surveys; GPS are measured using GNSS (GPS) technology, whereas EGM stands for the Earth Gravitational Model.

$$
\begin{bmatrix}
\boldsymbol{\varepsilon}_{\mathbf{TS6}} \\
\boldsymbol{\varepsilon}_{\mathbf{TS7}} \\
\boldsymbol{\varepsilon}_{\mathbf{TS8}} \\
\boldsymbol{\varepsilon}_{\mathbf{TS9}} \\
\boldsymbol{\varepsilon}_{\mathbf{TS10}} \\
\boldsymbol{\varepsilon}_{\mathbf{TS11}} \\
\boldsymbol{\varepsilon}_{\mathbf{TS12}} \\
\boldsymbol{\varepsilon}_{\mathbf{TS13}} \\
\hline
\boldsymbol{\varepsilon}_{\mathbf{GPS5}} \\
\boldsymbol{\varepsilon}_{\mathbf{GPS6}} \\
\boldsymbol{\varepsilon}_{\mathbf{GPS7}} \\
\hline
\boldsymbol{\varepsilon}_{\mathbf{EGM2}}
\end{bmatrix}
= \mathbf{X}
\begin{bmatrix}
\mathbf{d}_5 \\
\mathbf{d}_6 \\
\mathbf{d}_7 \\
\mathbf{d}_C \\
\mathbf{d}_D \\
\hline
d\xi_2 \\
d\eta_2 \\
d\Sigma_2
\end{bmatrix}
+ \mathbf{U}
\begin{bmatrix}
\mathbf{d}_1 \\
\mathbf{d}_2 \\
\mathbf{d}_3 \\
\mathbf{d}_4 \\
\mathbf{d}_A \\
\mathbf{d}_B \\
\hline
d\xi_1 \\
d\eta_1 \\
d\Sigma_1
\end{bmatrix}
- \mathbf{y}
\tag{43}
$$

where

- $\boldsymbol{\gamma} = \begin{bmatrix} \mathbf{d}_1^T & \mathbf{d}_2^T & \mathbf{d}_3^T & \mathbf{d}_4^T & \mathbf{d}_A^T & \mathbf{d}_B^T & | & d\xi_1 & d\eta_1 & d\Sigma_1 \end{bmatrix}^T$ is the vector of unknown corrections to all parameters adjusted on the first total station position, with zero expected value and a known covariance matrix $\boldsymbol{\Sigma}_{\boldsymbol{\gamma}} = \boldsymbol{\Sigma}_{\hat{\boldsymbol{\beta}}}$.

- $\boldsymbol{\beta} = \begin{bmatrix} \mathbf{d}_5^T & \mathbf{d}_6^T & \mathbf{d}_7^T & \mathbf{d}_C^T & \mathbf{d}_D^T & | & d\xi_2 & d\eta_2 & d\Sigma_2 \end{bmatrix}^T$ is the vector of corrections $dX, dY, dZ$ to the approximate geocentric coordinates $X, Y, Z$ of the new points 5, 6, 7, C, D included in the model at the second total station position, and corrections $d\xi_2, d\eta_2, d\Sigma_2$ to the total station orientation angles;

- $\boldsymbol{\varepsilon} = \begin{bmatrix} \boldsymbol{\varepsilon}_{TS6}^T & \boldsymbol{\varepsilon}_{TS7}^T & \cdots & \boldsymbol{\varepsilon}_{TS13}^T & | & \boldsymbol{\varepsilon}_{GPS5}^T & \boldsymbol{\varepsilon}_{GPS6}^T & \boldsymbol{\varepsilon}_{GPS7}^T & | & \boldsymbol{\varepsilon}_{EGM3}^T \end{bmatrix}^T$ is the vector of total station measurement random errors $\varepsilon_x, \varepsilon_y, \varepsilon_z$ of the $x, y, z$ coordinates of the points 1, 3, 4, 6, 7, 5, C, D, respectively; the GNSS measurement's random errors $\varepsilon_X, \varepsilon_Y, \varepsilon_Z$ of the $X, Y, Z$ coordinates of the points 5, 6, 7, respectively; and the random errors $\varepsilon_\xi, \varepsilon_\eta$ of the deflection of the vertical components $\xi_2, \eta_2$ at the second total station position (at point 2): $\boldsymbol{\varepsilon}_{EGM2}^T = \begin{bmatrix} \varepsilon_{\xi_2} & \varepsilon_{\eta_2} \end{bmatrix}$;

- $\mathbf{X}, \mathbf{y}$ is the known design matrix and the free terms vector.

The solution of the observational–mixed model at the second total station position $\boldsymbol{\varepsilon} = \mathbf{X}\boldsymbol{\beta} + \mathbf{U}\boldsymbol{\gamma} - \mathbf{y}$ is given by Equations (33)–(40).

At the third total station position over the point 5 (see Figures 2 and 5), the observational-mixed model $\boldsymbol{\varepsilon} = \mathbf{X}\boldsymbol{\beta} + \mathbf{U}\boldsymbol{\gamma} - \mathbf{y}$ is given by

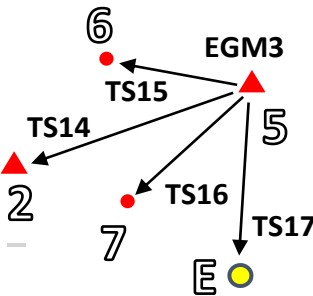

**Figure 5.** Surveying geometry at the total station position 5. The numbered points marked as TS relate to total station surveys, and EGM stands for the Earth Gravitational Model.

$$
\begin{bmatrix} \boldsymbol{\varepsilon}_{\mathbf{TS14}} \\ \boldsymbol{\varepsilon}_{\mathbf{TS15}} \\ \boldsymbol{\varepsilon}_{\mathbf{TS16}} \\ \boldsymbol{\varepsilon}_{\mathbf{TS17}} \\ \hline \boldsymbol{\varepsilon}_{\mathbf{EGM3}} \end{bmatrix} = \mathbf{X} \begin{bmatrix} \mathbf{d}_{\mathrm{E}} \\ \hline d\xi_3 \\ d\eta_3 \\ d\Sigma_3 \end{bmatrix} + \mathbf{U} \begin{bmatrix} \mathbf{d}_1 \\ \mathbf{d}_2 \\ \mathbf{d}_3 \\ \mathbf{d}_4 \\ \mathbf{d}_{\mathrm{A}} \\ \mathbf{d}_{\mathrm{B}} \\ \hline d\xi_1 \\ d\eta_1 \\ d\Sigma_1 \\ \hline \mathbf{d}_5 \\ \mathbf{d}_6 \\ \mathbf{d}_7 \\ \mathbf{d}_{\mathrm{C}} \\ \mathbf{d}_{\mathrm{D}} \\ \hline d\xi_2 \\ d\eta_2 \\ d\Sigma_2 \end{bmatrix} - \mathbf{y} \tag{44}
$$

where

- $\boldsymbol{\gamma} = [\mathbf{d}_1^T \quad \mathbf{d}_2^T \quad \mathbf{d}_3^T \quad \mathbf{d}_4^T \quad \mathbf{d}_{\mathrm{A}}^T \quad \mathbf{d}_{\mathrm{B}}^T \quad | \quad d\xi_1 \quad d\eta_1 \quad d\Sigma_1 \;\vdots\; \mathbf{d}_5^T \quad \mathbf{d}_6^T \quad \mathbf{d}_7^T \quad \mathbf{d}_{\mathrm{C}}^T \quad \mathbf{d}_{\mathrm{D}}^T \quad | \quad d\xi_2 \quad d\eta_2 \quad d\Sigma_2]^T$ is the vector of unknown corrections to all parameters adjusted on the second total station position, with zero expected value and a known covariance matrix:

$$
\boldsymbol{\Sigma_\gamma} = \begin{bmatrix} \boldsymbol{\Sigma}_{\hat{\beta}} & \boldsymbol{\Sigma}_{\hat{\beta}\hat{\gamma}} \\ \boldsymbol{\Sigma}_{\hat{\beta}\hat{\gamma}}^T & \boldsymbol{\Sigma}_{\hat{\gamma}} \end{bmatrix} \tag{45}
$$

- $\boldsymbol{\beta} = \begin{bmatrix} \mathbf{d}_{\mathrm{E}}^T & | & d\xi_3 & d\eta_3 & d\Sigma_3 \end{bmatrix}^T$ is the vector of corrections $dX, dY, dZ$ to the approximate geocentric coordinates $X, Y, Z$ of the new point E included in the model at the third total station position, and corrections $d\xi_3, d\eta_3, d\Sigma_3$ to the total station orientation angles;

- $\boldsymbol{\varepsilon} = \begin{bmatrix} \boldsymbol{\varepsilon}_{TS14}^T & \boldsymbol{\varepsilon}_{TS15}^T & \boldsymbol{\varepsilon}_{TS16}^T & \boldsymbol{\varepsilon}_{TS17}^T & | & \boldsymbol{\varepsilon}_{EGM3}^T \end{bmatrix}^T$ is the vector of total station measurement random errors $\varepsilon_x, \varepsilon_y, \varepsilon_z$ of the $x, y, z$ coordinates of the points 2, 6, 7, E, respectively; and random errors $\varepsilon_\xi, \varepsilon_\eta$ of the deflection of the vertical components $\xi_3, \eta_3$ at the third total station position (at point 5): $\boldsymbol{\varepsilon}_{EGM3}^T = \begin{bmatrix} \varepsilon_{\xi_3} & \varepsilon_{\eta_3} \end{bmatrix}$;

- $\mathbf{X}, \mathbf{y}$ is the known design matrix and free terms vector.

The solution of the observational–mixed model at the third total station position $\boldsymbol{\varepsilon} = \mathbf{X}\boldsymbol{\beta} + \mathbf{U}\boldsymbol{\gamma} - \mathbf{y}$ is given by Equations (33)–(40).

The results of the sequential online adjustments of the total station positions were compared to those of the aforementioned classical offline adjustment of the total station traverse, including the merging points, as presented in the paper [28]. The obtained extremal coordinates' differences are only $10^{-6}$ m. That technically means that the proposed method and algorithm of the sequential online adjustment of the total station positions work correctly, and give the same results as the classical offline adjustment of the traverse, including the merging points, as expected.

2.4.3. Sequential Online Adjustments of the Total Station Observations

At the first total station position, the coordinate vectors of the station 1-point **GPS1** $= \begin{bmatrix} X_1 & Y_1 & Z_1 \end{bmatrix}^T$ and the target 2-point **GPS2** $= \begin{bmatrix} X_2 & Y_2 & Z_2 \end{bmatrix}^T$ have to be determined via a GNSS receiver. These position vectors and the vertical deflection components vector **EGM1** $= \begin{bmatrix} \xi_1 & \eta_1 \end{bmatrix}^T$ are used for computation of the approximate value of the horizontal angle $\Sigma_1$. The observational mixed model $\varepsilon = \mathbf{X}\boldsymbol{\beta} + \mathbf{U}\boldsymbol{\gamma} - \mathbf{y}$ for the first total station observation **TS1** $= \begin{bmatrix} x_1 & y_1 & z_1 \end{bmatrix}^T$, including the GPS1, GPS2 and EGM1 data, is given by

$$\begin{bmatrix} \varepsilon_{\mathbf{TS1}} \\ \varepsilon_{\mathbf{GPS1}} \\ \varepsilon_{\mathbf{GPS2}} \\ \varepsilon_{\mathbf{EGM1}} \end{bmatrix} = \mathbf{X}[d\Sigma_1] + \mathbf{U} \begin{bmatrix} \mathbf{d}_1 \\ \mathbf{d}_2 \\ \hline d\xi_1 \\ d\eta_1 \end{bmatrix} - \mathbf{y} \tag{46}$$

where the covariance matrices $\boldsymbol{\Sigma}_\varepsilon, \boldsymbol{\Sigma}_\gamma$ of the vectors $\varepsilon, \gamma$ are known a priori.

The solution of the mixed model is given by following Equations (33)–(40).

For the second total station observation TS2, the mixed model is defined as

$$\varepsilon_{\mathbf{TS2}} = \mathbf{X}\mathbf{d}_3 + \mathbf{U} \begin{bmatrix} \mathbf{d}_1 \\ \mathbf{d}_2 \\ \hline d\xi_1 \\ d\eta_1 \\ d\Sigma_1 \end{bmatrix} - \mathbf{y} \tag{47}$$

where $\boldsymbol{\gamma} = \begin{bmatrix} \mathbf{d}_1^T & \mathbf{d}_2^T & | & d\xi_1 & d\eta_1 & d\Sigma_1 \end{bmatrix}^T$ is the vector of unknown corrections to all parameters adjusted previously, with zero expected value and a known covariance matrix:

$$\boldsymbol{\Sigma}_\gamma = \begin{bmatrix} \boldsymbol{\Sigma}_{\hat{\boldsymbol{\beta}}} & \boldsymbol{\Sigma}_{\hat{\boldsymbol{\beta}}\hat{\gamma}} \\ \boldsymbol{\Sigma}_{\hat{\boldsymbol{\beta}}\hat{\gamma}}{}^T & \boldsymbol{\Sigma}_{\hat{\gamma}} \end{bmatrix} \tag{48}$$

Its solution is given by Equations (33)–(40).

The next total station observation vectors TS4, TS5, . . . are included in the model and adjusted in the same way. The GPS3, GPS4, . . . vectors can be included and adjusted after measurements at any time. For example, if the GPS3 vector is included after adjustment of the second total station observation TS2, the mixed model does not contain the new parameters' vector $\beta$:

$$\varepsilon_{\mathbf{GPS3}} = \mathbf{U} \begin{bmatrix} \mathbf{d}_1 \\ \mathbf{d}_2 \\ \hline d\xi_1 \\ d\eta_1 \\ d\Sigma_1 \\ \hline \mathbf{d}_2 \end{bmatrix} - \mathbf{y} \tag{49}$$

where $\boldsymbol{\gamma} = \begin{bmatrix} \mathbf{d}_1^T & \mathbf{d}_2^T & \mathbf{d}_3^T & | & d\xi_1 & d\eta_1 & d\Sigma_1 \end{bmatrix}^T$ is the vector of unknown corrections to all parameters adjusted previously, with zero expected value and a known covariance matrix:

$$\boldsymbol{\Sigma}_{\boldsymbol{\gamma}} = \begin{bmatrix} \boldsymbol{\Sigma}_{\hat{\boldsymbol{\beta}}} & \boldsymbol{\Sigma}_{\hat{\boldsymbol{\beta}}\hat{\boldsymbol{\gamma}}} \\ \boldsymbol{\Sigma}_{\hat{\boldsymbol{\beta}}\hat{\boldsymbol{\gamma}}}^T & \boldsymbol{\Sigma}_{\hat{\boldsymbol{\gamma}}} \end{bmatrix} \tag{50}$$

After measuring and adjusting all vectors at the first total station position, for example, in the sequence {TS1+ GPS1+ GPS2+ EGM1}, TS2, GPS3, TS3, TS4, TS5, GPS4, the adjusted sequential process is continued on the second position of the total station. Firstly, the previously adjusted position vectors GPS1, GPS2 and vector of the vertical deflection components $\mathbf{EGM2} = \begin{bmatrix} \xi_2 & \eta_2 \end{bmatrix}^T$ are used for computing the approximate value of the horizontal angle $\boldsymbol{\Sigma}_2$. The observational–mixed model $\boldsymbol{\varepsilon} = \mathbf{X}\boldsymbol{\beta} + \mathbf{U}\boldsymbol{\gamma} - \mathbf{y}$ for the first total station observation on the second position $\mathbf{TS6} = \begin{bmatrix} x_1 & y_1 & z_1 \end{bmatrix}^T$ is defined by

$$\begin{bmatrix} \boldsymbol{\varepsilon}_{\mathbf{TS1}} \\ \boldsymbol{\varepsilon}_{\mathbf{EGM1}} \end{bmatrix} = \mathbf{X}[d\Sigma_2] + \mathbf{U} \begin{bmatrix} \mathbf{d}_1 \\ \mathbf{d}_2 \\ \hline d\xi_1 \\ d\eta_1 \\ d\Sigma_1 \\ \hline \mathbf{d}_3 \\ \mathbf{d}_4 \\ \mathbf{d}_A \\ \mathbf{d}_B \\ \hline d\xi_2 \\ d\eta_2 \end{bmatrix} - \mathbf{y} \tag{51}$$

where the covariance matrices $\boldsymbol{\Sigma}_{\boldsymbol{\varepsilon}}, \boldsymbol{\Sigma}_{\boldsymbol{\gamma}}$ of the vectors are known a priori.

The solution of the mixed model is provided by solving Equations (33)–(40). Next, data obtained from the measurements at station 2 are included in the model and adjusted in the same way as the first total station position.

The adjusted sequential process is continued on the third position of the total station in the same way.

The results of the sequential online adjustments of the total station observations were compared to the classical offline adjustment of the total station traverse mentioned earlier, excluding the merging points presented in the paper [28]. The obtained extremal coordinates' differences are only in the order of $10^{-6}$ m. This technically means that the proposed method and algorithm of the sequential online adjustments of the total station observations work correctly, and give the same results as the classical offline adjustment of the traverse without merging points, as expected.

## 3. Discussion

In our studies, we have presented the algorithm for real-time adjustment of the integrated GNSS, total station, and EGM vertical direction data in the geocentric coordinate system. The proposed method has two modes: sequential online adjustment of the total station positions and online adjustment of the total station observations. Both modes have been tested on data from the spatial total station traverse measurements. In order to verify the correctness of the online algorithm, the total station traverse data were also adjusted using the offline algorithm. The offline adjustment was conducted with and without including the merging points.

Online and offline adjustment computation should be considered complementary approaches. They are an essential aspect of the assessment of surveying results, especially

in issues related to displacement measurements and deformation monitoring. As demonstrated in the previous section, the accuracies offered by both procedures are comparable and utterly applicable to field measurements. Nevertheless, the main advantages of online adjustment are the ability to conduct reliable deformation monitoring and the ongoing fulfillment of the obtained measurement results for risk management purposes. In this context, attention should be paid to the problem of notifying users of risks arising from the construction and exploitation of the monitored object, as well as from various natural phenomena affecting changes in its structural geometry. Defining safety thresholds (so-called "limit checks") is not straightforward. This is primarily caused by the need to correctly identify a specific object and its surroundings, which in turn requires a reasonably long observation period and using multi-source data. Its integration in a standardized manner is one of the most significant theoretical challenges posed by modern geodesy.

Moreover, ongoing technological advancement influences the constant modification of once-accepted assumptions. As a rule, data sets representing surveying results captured continuously or quasi-continuously onsite must reliably identify outlier observations and determine the influence of other error sources, be they systematic, personal, or instrumental. To this end, complementary methods using estimation and optimization procedures are becoming increasingly important. As discussed in our paper, such a need to work online encourages the development of appropriate procedures and algorithms.

As demonstrated in the conducted studies, including information regarding the local distribution of plumb line parameters in total station surveying is expedient and desirable. To this end, total station and GNSS satellite measurements were concatenated within the EGM2008 geoid model. Such an approach unifies operating surveying instruments, driving them to be used precisely and universally. However, in the general assessment process, one should consider the specifics of the developed test network. After all, the object reveals specific features (urban, intensively built-up, and used areas) and geometrical features; the network is regular, set up following the art of measurement. We realize that depending on the altering conditions of surveys, the results may somewhat differ from those obtained in our experiment.

Nevertheless, as already shown in the literature review, similar work has been carried out in other conditions—for example, in the mountains—giving similar results. Additionally, incorporating further data sources into the observational system should not pose significant concerns. In such a case, the vectors of unknowns and the corresponding coefficient matrices will be modified, generally increasing the size of the corresponding arrays. Theoretically, this will cause increased demand for computer processing power; however, with the current parameters of available machines and computational optimization methods, this should be a manageable challenge.

## 4. Conclusions

In conclusion, we have formulated some closing statements and tracked the progress of further research in relevant fields. First, it should be noted that despite the many studies conducted worldwide on the integration and standardization of spatial data, the issue of effective data fusion still needs to be solved. This will mainly be achieved by the continuous development of measurement methods and surveying instruments, as well as by incorporating multi-source data into newly constructed surveying systems. Examples include numerous attempts to combine spatial (geometric) data with physical data. Secondly, scientific progress includes working on new geoid models and continuously improving existing ones. Such studies transform into continuous precision and accuracy validation of local plumb line parameters within the EGM framework. Undoubtedly, this requires ongoing test work using current data sources.

The results of our study can be summarized as follows:

- The total station traverse is very well spatially oriented by using the total station vertical axe deflection from normal to GRS 80 ellipsoid components obtained from EGM, and by measuring the coordinates of the total station standpoint using a GNSS

receiver. There is no need to carry out GNSS measurements at any additional merging total station traverse points.

- The proposed methods and algorithms of the sequential online adjustment of the total station positions and the sequential adjustment of the single total station vector of observations, including the GNSS and EGM data in both cases, work correctly. They deliver the same results as the classical offline adjustment of the total station traverse.

The proposed algorithms allow for real-time surveying with the online adjustment of the observations in the geocentric coordinate system, providing current control of the adjustment results. Our online computation methodology, applied to the discussed surveys (total station/GNSS/EGM), can be used to directly model 3D objects in real time with high spatial accuracy in a geocentric coordinate system. It can also be applied to measure control points for further use in close-range photogrammetry (aerial and terrestrial). The procedure can be applied while defining reliable tie points for merging point clouds in laser scanning (both terrestrial and performed using UAVs). Such an approach is beneficial in BIM while constructing and updating relevant 3D data models.

As our survey network was near the GNSS reference station (at a distance not exceeding 250 m), we also acknowledge the need to test the solution for other-longer distances. Our future works also encompass testing and improving new, multi-source data algorithms for object diagnostic purposes.

**Author Contributions:** Conceptualization, K.K. and E.O.; methodology, E.O. and K.K.; validation, K.K. and Z.M.; formal analysis, E.O., K.K. and Z.M.; investigation, E.O., K.K. and Z.M.; writing—original draft preparation, K.K.; writing—review and editing, K.K. and E.O.; supervision, E.O.; project administration, K.K. All authors have read and agreed to the published version of the manuscript.

**Funding:** This research received no external funding.

**Institutional Review Board Statement:** Not applicable.

**Informed Consent Statement:** Not applicable.

**Data Availability Statement:** The data presented in this study are available upon request from the co-author Edward Osada: edward.osada@dsw.edu.pl.

**Conflicts of Interest:** The authors declare no conflict of interest.

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
