# Peer review of "Real-Time Adjustment and Spatial Data Integration Algorithms Combining Total Station and GNSS Surveys with an Earth Gravity Model"

_applsci, doi:10.3390/app13169380_

Round 1

Reviewer 1 Report

Lines 33-34. Differences between the results of processing networks of different configurations do not indicate the high efficiency of the algorithms. This convergence only means that the results of the numerical experiment do not differ significantly from each other. Suggestion needs to be edited.

Line 71. The millimeter accuracy of the survey results must be proved on field standards. Comparison of calculation results does not indicate the accuracy of measurements. The proposal needs to be edited.

Line 77. A link is given to the short article, but not to the dissertation. Edit offer.

Lines 94-104. It is not disclosed how the approach to joint adjustment in the local coordinate system fundamentally differs from the author's approach. Why don't the authors of the article use a local topocentric coordinate system that is not related to the global ellipsoid? In our opinion, it is easier to switch from global GNSS coordinates to local ones. Provide an explanation for this in the text.

Line 117. Should disclose what the optimization is. Is it a mathematical procedure?

Fig.2. Indicate at which points GNSS measurements are performed.

Line 341. It should be indicated to which GNSS measurement mode (static, RTK or so) this accuracy of determining spatial coordinates belongs.

Line 341-345. It should be clarified what kind of differential method is meant. Code differential measurements have an accuracy of the order of decimeters. It's probably real-time kinematics. Write more precisely.

Line 345-346. In conclusions about the possibilities of the proposed approach, it should be clarified that all the results were obtained at a distance of no more than 250 m from the base station.

In Table 1, plumb line deviations are of the order of 0.01 arcseconds. Shouldn't they be taken the same for all stations? What gives this differentiation?

Line 376. Clarify what is meant by spatial stabilization of sites. It's some kind of jargon.

Line 450. It is necessary to clarify how the differences of 6-10 m indicate that the results obtained are the same. Correct or edit the offer.

Section 2.3.2. It is imperative to explain why the so-called offline algorithm cannot be applied in near real time. What prevents this? What software implements the proposed online processing? Describe it.

Lines 579-582. The offer contains unrealized intentions and is of an advertising nature. It has nothing to do with the content of the article. It should be removed.

Author Response

Dear Reviewer,

Attached, please find our responses to your valuable comments. 

We want to thank you very much for your effort.

Kind regards,

The authors

Reviewer 2 Report

Dear Colleagues,

I commend you on the quality of your paper; it is well-written, interesting, and practical. While I have come across similar ideas in other studies regarding the estimation of transformation parameters, such as determining earth orientation parameters using space geodetic measurements, I truly appreciate your unique approach and the method presented in this paper. However, to further enhance the rigor of your work, I recommend a moderate revision, and I have provided an annotated file for your reference. My comments are as follows:

1. Some sections of the paper lack clarity, and I have highlighted these points in the annotated file to facilitate better understanding.

2. It is essential to ensure consistency in mathematical notation throughout the text and all formulae. For instance, the usage of "X" for both coordinates and the design matrix, and "beta" as both a vector and an angle, can be confusing. I suggest adhering to standard mathematical presentation conventions: represent all matrices with capital letters in bold and non-italic font, all vectors with small letters in bold and non-italic font, and all other variables with small letters in italic font. Additionally, sub- or superscripts that are not variables should not be italicized, and matrix operators and mathematical functions should be in non-italic font. These conventions are widely accepted in mathematical presentations.

3. I recommend presenting tables to display your computation results, including the success of your estimations, uncertainties of the estimates, and statistical analyses. This will provide a clear overview of your findings and add depth to your research.

4. You have utilized EGM2008 for computing the deflections of the vertical, but you did not compute the errors associated with them. It would be beneficial to explain your rationale behind this decision, providing insight into the impact and accuracy of your chosen approach.

5. An important consideration when applying EGMs is the presence of shifts and drifts when comparing them with GNSS leveling data. I suggest referring to the relevant papers mentioned below to address this issue effectively and discuss how your study aligns with the observations made in these references:

Eshagh M. and Berntsson J. (2019) On quality of NKG2015 geoid model over the Nordic countries, Journal of Geodetic Science, 9, 97-110.  https://doi.org/10.1515/jogs-2019-0010

 Eshagh M. and Zoghi S. (2016) Local error calibration of EGM08 geoid using GNSS/levelling data, Journal of Applied Geophysics 130:209-217. 

Eshagh M. and Ebadi S. (2014) A strategy to calibrate errors of Earth gravity models, J Appl. Geophys. 103:215-220.

 Eshagh M. (2013) Numerical aspects of EGM08-based geoid computations in Fennoscandia regarding the applied reference Surface and error propagation, J Appl. Geophys, 96: 28-32. 

 Eshagh M. (2013) On the reliability and error calibration of some recent Earth's gravity models of GOCE with respect to EGM08, Acta Geod. Geophys. Hung., 48(2): 199-208.

Eshagh M. (2010) Error calibration of quasi-geoid, normal and ellipsoidal heights of Sweden using variance component estimation, Contr. Geophys. Geod., 40(1):1-30.

Kiamehr R. and Eshagh M. (2008) Estimation of variance components Ellipsoidal, Geoidal and orthometrical heights, J Earth & Space Phys., 34(3):1-13.

In all of these articles, the discussion revolves around the distance between the geoid and ellipsoid. In your study, the total station is situated at the Earth's surface, and the deflection of the vertical you used is the geoidal deflection of vertical, as I understand it. However, it appears that you needed to use the surface deflection of vertical, which involves the geoid height and introduces all the problems discussed in these articles regarding the differences between the reference ellipsoid and geoid surfaces. I strongly advise you to address this crucial issue in your paper."

6. I believe that the part related to the sequential approach should be improved. I have written comments in the annotated file to further elaborate on my suggestion. For example, you can see what I mean about a sequential adjustment procedure. I recommend referring to the following paper to gain a better understanding of my idea: [Insert paper citation here]. This paper will provide valuable insights into how a sequential adjustment procedure can be effectively applied and will assist in refining your approach in your study."

I hope these revisions accurately reflect your intended points and provide clarity to the readers of your comments. If you have any further feedback or additions you would like to include, please let me know, and I will be happy to assist further.

Eshagh M. (2011) Sequential Tikhonov Regularization: an alternative way for inverting satellite gradiometric data, ZfV., 136:113-121. Download

I trust that you will find my comments both instructive and valuable. I am eagerly looking forward to reviewing your revised paper and witnessing the progress made based on the feedback provided.

Best regards

Mehdi Eshagh.

Author Response

(The authors gave the same response as above.)

Round 2

Reviewer 2 Report

Dear Colleagues,

Thank you very much for your collaborartion. I checked your responses and the revised paper. Few very minor issues that you can fix it later in the proofreading.

The parameters of Eqs. (1), (2) and (3) should be italic.

Line 407 and 409, Please write Section, capital letter for S when you address the section by its number.

Thank you again and I wish you good luck with your paper.

Best regards

Mehdi Eshagh.